# Multiply Interpenetrating Polymer Networks: Preparation, Mechanical Properties, and Applications

**DOI:** 10.3390/gels5030036

**Published:** 2019-07-08

**Authors:** Panayiota A. Panteli, Costas S. Patrickios

**Affiliations:** Department of Chemistry, University of Cyprus, P. O. Box 20537, 1678 Nicosia, Cyprus

**Keywords:** triple networks, quadruple networks, quintuple networks, interpenetrating networks, hydrogels, mechanical properties

## Abstract

This review summarizes work done on triply, or higher, interpenetrating polymer network materials prepared in order to widen the properties of double polymer network hydrogels (DN), doubly interpenetrating polymer networks with enhanced mechanical properties. The review will show that introduction of a third, or fourth, polymeric component in the DNs would further enhance the mechanical properties of the resulting materials, but may also introduce other useful functionalities, including electrical conductivity, low-friction coefficients, and (bio)degradability.

## 1. Introduction

Polymer network hydrogels comprise cross-linked hydrophilic polymers which can swell to a very large extent in aqueous media, and, therefore, contain a high percentage of water. Thus, hydrogels possess both solid-like and liquid-like characteristics [1], and, consequently, are very similar to biological tissues. Their high water percentage renders hydrogels biocompatible and appropriate for use in the biomedical and technology fields, in applications including drug delivery, tissue engineering, soft contact lenses, water and blood retention, sensors, and actuators [2,3,4,5]. However, the high water content also downgrades the mechanical performance of hydrogels, conferring upon them a low Young’s modulus and brittle behavior [6], with a negative impact on their applications.

In the last two decades, several new hydrogel structures possessing improved mechanical properties have been reported. These include the topological/slide ring (SR) gels [7], the nanocomposite (NC) gels [8], and the double-network (DN) hydrogels [9]. The DN hydrogels consist of two interpenetrating polymer networks, the first one being brittle, comprising densely cross-linked polyelectrolyte chains, and the second being ductile, comprising loosely cross-linked nonionic polymer chains [9]. These DN hydrogels exhibit outstanding mechanical properties, much better than the sum of the mechanical properties of their two network components [10,11]. Furthermore, the preparation of DN hydrogels is very easy, much easier than that of the SR and the NC gels. Recently, the DN concept has been extended, leading to the preparation of triple-network (TN) hydrogels with extraordinary mechanical performance. Moreover, due the universality of this procedure, a great variety of monomers and cross-linkers can be employed, resulting in the preparation of TN hydrogels [12,13,14,15,16,17,18,19,20,21,22,23,24,25,26] with unique physical or chemical properties. Very recently, the DN concept has been further extended by preparing quadruple-network (QN) hydrogels [27,28] and quintuple-network (5×N) hydrogels [29,30], which also possess enhanced mechanical properties.

The mechanical behavior and the physical and chemical properties of these multiply interpenetrating polymer network hydrogels can be controlled by several parameters. These parameters include the number of polymer networks, the monomer type (influencing mechanical properties mainly through its effect on the degree of swelling) and concentration, and the cross-linking density in the first and in the higher networks. This review summarizes the literature work in this niche, but rapidly growing, area, including the design, preparation, characterization, and applications of the TN, QN and 5×N hydrogels developed until today.

## 2. Network Structures with Improved Mechanical Properties

### 2.1. Double-Network (DN) Hydrogels

The preparation of double-network (DN) hydrogels was reported in 2003 by Gong, Osada and co-workers [9]. DN hydrogels are based on two interpenetrating polymeric hydrogels with large differences in their mechanical properties, cross-linking density, and electric charge. In particular, the first network is brittle, consisting of a relatively highly cross-linked polyelectrolyte network of poly(2-acrylamido-2-methylpropane sulfonic acid) (PAMPS), containing 4 mol%, relative to the AMPS monomer, of *N*,*N*′-methylenebisacrylamide (MBAAm) cross-linker, while the second network is ductile, based on a relatively loosely cross-linked polyacrylamide (PAAm) nonionic network, containing 0.1 mol% MBAAm cross-linker relative to the monomer. These DN hydrogels are prepared in two steps. In the first step, the PAMPS single-network (SN) hydrogel is prepared. In the second step, this SN hydrogel is allowed to reach swelling equilibrium in an aqueous solution of 2 M acrylamide (AAm) monomer, also containing MBAAm cross-linker and 2-oxoglutaric acid (OXG) photoinitiator, followed by the subsequent photopolymerization of the AAm-MBAAm solution absorbed within the PAMPS first network, resulting in the formation of the second network and the final DN composite network. After their preparation, the mechanical properties of the SN and the DN hydrogels are evaluated using compression experiments. The PAMPS/PAAm DN hydrogels exhibit extraordinary mechanical performance in comparison with the PAAm and PAMPS SN hydrogels, in particular, achieving values of compressive fracture stress, *σ*_max_, of 17.2 MPa, compressive fracture strain, *ε*_max_, of 92%, and compressive fracture energy of up to ~300 J·m^‒2^, despite their high water content (~90 wt%) [9]. Figure 1 illustrates the structure of the two different polymer networks in the DN hydrogel (Figure 1A), and a photograph of a tough DN hydrogel containing 90 wt% water (Figure 1B).

### 2.2. Toughening Mechanism in DN Hydrogels

#### 2.2.1. Role of Each Network in DN Toughening

The main network component in classical DN hydrogels is the second network, as the polymer volume fraction from that network is about 10 times that originating from the first. Inhomogeneities in the first network are responsible for large energy dissipation in the DN via the early and irreversible fragmentation (crack formation) of this first network into smaller network pieces (microgels), whereas the presence of the second network, in particular, at a relatively high polymer volume fraction, prevents crack propagation in the overall DN, and holds together the broken pieces of the first network, consequently ensuring DN continuity on the macroscopic level.

#### 2.2.2. Interaction between the Two Network Components

Although it was originally proposed [9] that, unlike in typical interpenetrating polymer networks, the two network components in DNs do not interact with each other either attractively or repulsively, experimental evidence, subsequently compiled, indicated the contrary, and revealed attractions, both physical and chemical [10]. Regarding the former, small-angle neutron scattering data suggested that the two networks are (enthalpically) attracted to each other, at least more so than each network is attracted to water. Regarding the latter, chemical bonding was identified between the two networks. This network covalent interconnectivity was proven as crucial for securing force transmission between the two network components, and attaining optimal mechanical properties. Interconnectivity usually takes place automatically via unsaturations within the first network arising from singly-reacted divinyl cross-linker molecules [31], and onto which chains of the second network are grafted. Elimination of this interconnectivity leads to a deterioration of the mechanical properties of the DN, unless the chains in the second network are highly entangled with those of the first network (as well as with themselves, i.e., the other chains in the second network), afforded with the use of a very small cross-linker loading during the preparation of the second network.

#### 2.2.3. Toughening Mechanism in DN Hydrogels

Their complex polymeric structure and resultingly enhanced mechanical properties are the reason why DN hydrogels do not follow the Lake–Thomas fracture model, which well describes the fracture of common (single) polymeric hydrogels. Furthermore, the enhancement of the mechanical properties of DN hydrogels is not described well by mechanisms relevant to the strengthening of other polymeric materials, including viscous dissipation and bulk viscoelastic losses. Instead, a ***local yielding*** mechanism seems to be in action, sharing common features with the fracture mechanisms of other tough materials, including rubbers, glassy polymers, bones and metals. According to this mechanism, a thin damaged zone is formed as a result of the cracking (breaking) of the first (brittle) polymer network which is now converted to smaller, discontinuous fragments [10]. These fragments are held together by the second (ductile) polymer network, thereby securing the continuity of the complex material.

### 2.3. Triple-Network (TN) Hydrogels

This is the main section of the review, and it will present the developed triple-network (TN) hydrogels. Many of these TN hydrogels comprise classical DN hydrogels plus a third polymer network. In several cases, the aforementioned DN hydrogel is the classical PAMPS/PAAm DN hydrogel developed by Gong, Osada and co-workers. In some other cases, the third component is not a polymer network, but rather a linear (not cross-linked) polymer, or some other polymer architecture, such as a polymeric microgel. This non-cross-linked polymer entity may be the first, the second, or the third component of the TN hydrogels. Thus, the presented TN hydrogels may not necessarily comprise three polymer networks, but just two, with the third component being physically entrapped within the overall network structure.

In cases where TN hydrogels are not partially composed of classical DN hydrogels, and do not bear a polyelectrolyte component, they lack Coulombic repulsions among the polymer network chains, and lack counterions whose translational entropy would add to the osmotic pressure of the system. This lack of repulsions and extra osmotic pressure lead to reduced equilibrium network swelling, which would allow the introduction of only a small volume of solution of monomer necessary for the preparation of the next polymer network or polymeric component, resulting only in a limited enhancement of the mechanical properties. This undesired situation of low swelling can be faced in TN hydrogels, if the second and third networks are prepared using a solution containing the monomer at a high concentration and the cross-linker at a very low, or even zero, concentration. In this case, monomer polymerization during the preparation of the second and third polymer network (or polymeric component) would result in a large loss of translational entropy, without the introduction of many (or any at all) new cross-links, imparting upon the newly-formed polymer component and the overall complex polymer network a relatively large swelling capacity, important for subsequent mechanical reinforcement. This crucial concept of exploiting the polymerization-induced loss of translational entropy to make up for absence of charge was first shown by Okay and co-workers [16] and Creton and co-workers [17] in TN hydrogels and TN elastomers, respectively.

The mechanism of the enhancement of the mechanical properties of the TN hydrogels, and, in general, of the multiple network hydrogels, is expected to be similar to that of the DN hydrogels. In particular, the earlier networks, which are more strained, should fracture first and dissipate mechanical energy, especially when they contain large heterogeneities [16], while the later networks should hold together the fragments of the damaged networks, and delay crack propagation. Furthermore, these later networks should increase the thickness of the damaged zone to enhance the dissipated energy upon crack formation. This may imply a possible optimal design for multiple network hydrogels, in which each subsequent network comprises a gradually higher monomer concentration and a gradually lower cross-linker concentration.

Table 1 summarizes the composition and mechanical properties of the TN hydrogels reported in the literature. The three first columns in the table contain the polymeric components of the TN hydrogels, and, in particular, the names of the monomers (or polymers) used and their concentration, as well as the name and the concentration of the cross-linker. The chemical structures and names of the monomers employed in the preparation of the TN hydrogels are presented in Table 2. In some cases, the TN hydrogels were cross-linked through physical interactions, hence these examples will be referred to as “physically cross-linked” (PC). The following columns contain the type of mechanical testing, whether compressive or tensile, and the mechanical properties of the SN, DN and TN hydrogels, and, in particular, the fracture stress (*σ*_max_, in MPa), the fracture strain (*ε*_max_, in %), and the Young’s modulus (*Ε*, also in MPa).

#### 2.3.1. Simple TN Hydrogels

In 2005, the research group of Gong [12] reported the synthesis of two new materials with enhanced mechanical properties and low frictional coefficients, prepared by introducing a third component in the already developed PAMPS/PAAm DN hydrogels [9], either a loosely cross-linked PAMPS network to prepare a triple-network (TN) hydrogel, or a linear PAMPS chain to prepare a TN hydrogel, denoted as DN-L. Thus, the TN hydrogels consisted of a highly cross-linked PAMPS network as the first network, a loosely cross-linked PAAm network as the second network, and a loosely cross-linked PAMPS polymer network or non-cross-linked PAMPS linear polymer chains as the third component. The thus-prepared DN, TN and DN-L hydrogels were characterized in terms of their mechanical properties using compression experiments which indicated a substantial increase in the Young’s modulus values of the TN and DN-L (~2 MPa), a slight increase in the fracture stress value of the TN (4.8 MPa), and a two-fold increase in the fracture stress of the DN-L hydrogel (9.2 MPa) in comparison with the DN hydrogel (*σ*_max_ = 4.6 MPa, *E* = 0.84 MPa), despite their identical water content. These increases in the Young’s moduli were attributed to the presence of the third polymer network, PAMPS, and not in the presence or absence of the MBAAm cross-linker. The enhancement of the fracture stress value of the DN-L hydrogel was attributed to the absence of the MBAAm cross-linker in the third network, leading to a linear PAMPS chain that can dissipate fracture energy more efficiently than its loosely cross-linked network counterpart. Finally, the DN-L hydrogel exhibited an ultra-low frictional coefficient (~10^−5^) against a glass substrate, which, together with its high mechanical toughness, renders this hydrogel an important candidate as material for artificial articular cartilage.

Three years later, the same research group [13] prepared TN hydrogels with increased mechanical toughness that were also investigated in terms of their ability to promote cell spreading and proliferation using three kinds of cells, bovine fetal aorta endothelial cells (BFAECs), human umbilical vein endothelial cells (HUVECs), and rabbit synovial tissue-derived fibroblast cells (RSTFCs). The first network consisted of a densely cross-linked 2-poly(acrylamido-2-methyl-propane sulfonic acid sodium salt) (PSAMPS) network, the second network consisted of a loosely cross-linked poly(*N*,*N*-dimethylacrylamide) (PDMAAm) network, and the third network consisted of an equimolar random copolymer of DMAAm and SAMPS, randomly cross-linked with 0, 2, and 4 mol% of MBAAm cross-linker. The resulting SN, DN, and TN hydrogels were then characterized in terms of their mechanical performance using compression experiments. The TN hydrogels exhibited improved mechanical properties in comparison with their SN counterparts, and, in particular, had fracture stress values of 1–3 MPa, fracture strain values of 47–71%, and Young’s modulus values of 0.32–0.88 MPa. Increasing the MBAAm cross-linker loading in the third polymer network resulted in increased Young’s modulus values and decreased values of fracture stress and fracture strain. Finally, the TN hydrogels were evaluated in terms of their ability to serve as a matrix for cell spreading and proliferation, as the presence of the ionic component SAMPS on the surface of the gel is known to promote cell adhesion and proliferation. It was found that cell proliferation was only observed in the cases of the TN hydrogels containing MBAAm cross-linker in the third polymer network, as the TN hydrogel consisting of linear polymer chains as the third polymeric component did not exhibit cell proliferation, because the linear polymer chains do not facilitate cell spreading due to their increased mobility.

In 2011, Zhang et al. [14,15] reported the preparation of injectable triply interpenetrating network hydrogels [14] and monodispersed spherical microgels with a triply interpenetrating structure [15], both containing three different natural components: partially oxidized dextran (Odex) prepared by oxidation of dextran using sodium periodate (NaIO_4_), *N*-carboxyethyl chitosan (CEC) synthesized from the reaction of chitosan and acrylic acid (AAc), and Teleostean. The aqueous solutions of the three natural components at varying polymer concentrations were mixed together in an aqueous buffer solution of pH 7.4 in the order: first Odex, then Teleostean, and finally CEC at a constant Odex:Teleostean:CEC ratio of 2:1:1 in order to form the triply interpenetrating hydrogels. Figure 2 illustrates the chemical structures of the three components employed in the preparation of these hydrogels. CEC is an amphoteric natural polymer as it contains both ‒NH_2_ and ‒CO_2_H groups, Teleostean contains ‒NH_2_ groups, and Odex contains –CHO groups. Thus, Odex acts as a macromolecular cross-linker both for CEC and Teleostean through imine (Schiff base) bond formation. Hence, the triply interpenetrating hydrogels are cross-linked through both covalent Schiff base bonds and ionic interactions between the amine and the carboxylic acid groups. After their preparation, these hydrogels were evaluated in terms of their mechanical properties using burst strength testing. The triple hydrogels exhibited higher mechanical performance than their doubly interpenetrating hydrogel counterparts. Furthermore, increasing the concentration of the constituents led to increased mechanical performance of the final hydrogels, as increasing the number of the ‒NH_2_, ‒CO_2_H, and ‒CHO groups resulted in higher cross-linking densities.

In 2014, Okay and co-workers [16] developed nonionic TN hydrogels based on the hydrophilic AAm or DMAAm monomers and the poly(ethylene glycol) dimethacrylate (PEGDMA) cross-linker, prepared using redox-initiated polymerization, with ammonium persulfate (APS) serving as the initiator and *N*,*N*,*N*′,*N*′-tetramethylethylenediamine (TEMED) as the polymerization accelerator. The PAAm/PAAm/PAAm TN hydrogels consisted of an inhomogeneous, highly cross-linked first polymer network of AAm, with a PEGDMA cross-linker loading up to 4 mol% with respect to monomer, and linear, uncross-linked PAAm chains as the second and third TN components. Similarly to the PAAm-based TN hydrogels, the PDMAAm/PDMAAm/PDMAAm TN hydrogels also consisted of an inhomogeneous highly cross-linked first polymer network of DMAAm, but with a PEGDMA cross-linker loading ranging between 4–10 mol% with respect to monomer, and linear PDMAAm chains as the second and third TN components. In contrast to the AAm or DMAAm concentration in the first network, which was constant, the AAm or DMAAm concentration in the second and third polymer networks acquired values between 0.7 and 7 M or 0.5 and 5 M, respectively, so as to obtain DN and TN hydrogels with varying molar ratios of the second to the first network units, *n*_21_, and second and third to the first network units, *n*_32/1_, respectively, thereby leading to the preparation of a large number of DN and TN hydrogels. All the hydrogels were evaluated in terms of their mechanical behavior using compression experiments. Figure 3 presents typical stress–strain curves for particular DN and TN hydrogels as the dependence of the nominal fracture stress (*σ*_nom_, solid curves) and true fracture stress (σ_true_, dashed curves, *σ*_true_ = λ × σ_nom_) on the deformation ratio, λ. Both the PAAm-based and PDMAAm-based TN hydrogels exhibited excellent mechanical properties, with fracture stress values of 25 and 26 MPa, respectively, fracture strain values of 99% and 95%, respectively, and Young’s modulus values of up to 2 MPa. Increasing the molar ratio of the repeating units in the higher networks to the repeating units in the first network, *n*_21_ and *n*_32/1_, resulted in a greater enhancement of the mechanical properties of the DN and TN hydrogels, respectively. Finally, cyclic compressive tests on a particular TN hydrogel revealed the presence of hysteresis in the first cycle due to permanent internal fractures taking place in the first network, and a nearly elastic behavior in the subsequent cycles due to the presence of the ductile PAAm or PDMAAm components that prevented the sample from failure.

At about the same time as Okay et al. [16], Creton et al. [17] prepared TN elastomers (as opposed to TN hydrogels) comprising the hydrophobic (water-insoluble) monomers ethyl acrylate (EA, always making up the first network, and, in some cases, the second and third networks too) and / or methyl acrylate (MA, mostly making up the second and third networks), and the hydrophobic cross-linker butanediol diacrylate (BDA). The concentration of EA monomer in the first network was always constant and equal to 5 M, while the concentration of BDA cross-linker in the first network acquired values of 1.45 (denoted with 0.5), 2.81 (denoted with 1), and 5.81 mol% (denoted with 2) with respect to monomer. In the higher networks, the concentration of EA and MA was 9.4 and 11 M, respectively, while the concentration of the BDA cross-linker was very low, and, in particular, 0.01 mol% relative to monomer. The prepared TN elastomers were subsequently evaluated in terms of their mechanical performance using tensile measurements, which revealed the superiority of these materials relative to their DN elastomer precursors. In particular, the fracture stress of the TN elastomers was found to be up to 29 MPa, four times higher than that of the DN elastomers, and the Young’s modulus was 4.2 MPa, three times higher than that of the DN elastomers. Figure 4 presents the true stress–stretch curves for the EA_1_SN, EA_1_MA DN, EA_1_MAMA TN, and the PMA second network alone at 60 °C (45 °C above the *T*_g_ of PMA), while Figure 4b presents the effect of cross-linker concentration in the EA first network on the EA_x_MA DN elastomers. Increasing the network’s multiplicity resulted in increased values of fracture stress, while decreasing the BDA cross-linker loading in the first network led to increased values of both fracture stress and fracture strain of the DN elastomers. Similar to the DN hydrogels, the origin of the toughening mechanism in the TN elastomer toughening was apparently due to prevention of crack formation and propagation in the first network by the higher (second and third) networks, in combination with the stiffening of the first network as limited by the fracture of its covalent bonds. Covalent bond fracturing in the first network was followed optically through the light emitted by a chemiluminescent cross-linker incorporated in the first network. Indeed, the cumulative mechanical hysteresis in the system was found to be directly proportional to the total chemiluminescence intensity, as shown in Figure 5.

In 2015, the research group of Okay [18] prepared DN and TN hydrogels based on methacrylated hyaluronan (MHA) macromonomer, DMAAm monomer, and MBAAm cross-linker, using sequential free radical photopolymerizations. In particular, these TN hydrogels consisted of poly(methacrylated hyaluronan) (PMHA) as the first network, and two loosely cross-linked polymer networks of PDMAAm as the second and third networks. A great variety of SN, DN, and TN hydrogels were prepared, afforded by the variation of the methacrylation degree in hyaluronan from 4% to 25%, the concentration of DMAAm monomer in the second network from 1 to 5 M and in the third network from 1 to 3 M, and the ratio of the MBAAm cross-linker to the DMAAm monomer in the second and third networks from 0.00–0.45 mol%. After their preparation and equilibrium swelling in water, all the prepared hydrogels were characterized in terms of their mechanical properties using compression experiments. The optimum methacrylation degree in MHA and the optimum ratio of MBAAm cross-linker to DMAAm monomer were found to be equal to 4% and 0.05 mol%, respectively. Similarly to the previous works by Okay and co-workers [26], the key factor for obtaining tough hydrogels was the molar ratio of the repeating units in the higher networks to the repeating units in the first network. Thus, increasing this ratio led to a greater enhancement of the mechanical properties. The toughest DN hydrogel presented a fracture stress value of 12 MPa, a fracture strain of 93%, and a Young’s modulus of 0.37 MPa. In the case of the TN hydrogels, the best-performing hydrogel exhibited a fracture stress of 22 MPa, a fracture strain of 95%, and a Young’s modulus of 0.4 MPa. Finally, the cyclic compressive test on a particular TN hydrogel indicated the same behavior as the behavior of a particular PAAm/PAAm/PAAm TN hydrogel, that exhibited hysteresis in the first cycle and an elastic behavior in the subsequent cycles.

#### 2.3.2. TN Hydrogels with Microgels as the First Component

In 2011, the research group of Gong [19,20] reported the preparation of TN hydrogels consisting of microgels as the first component, and loosely cross-linked PAAm networks as the second and third components. Five different monomers were employed in the preparation of the microgels. These were AAm, SAMPS, sodium 4-styrenesulfonate (NaSS), 2-(trimethylamino)ethyl acrylate, chloride quaternary salt (DMAEA-Q), and 3-(acrylamidopropyl)-trimethylammonium chloride (DMAPAA-Q). For each monomer, a different microgel was prepared, and, from that, the corresponding DN and TN hydrogels were formed. The preparation of the five different TN hydrogels was accomplished through three steps. Figure 6 illustrates the synthetic route for the preparation of the PSAMPS/PAAm/PAAm TN hydrogels. After their preparation, the microgel precursors were allowed to reach swelling equilibrium in an aqueous solution of AAm, containing MBAAm and OXG photoinitiator, and after their photopolymerization, they were again allowed to reach swelling equilibrium in an aqueous solution of AAm, MBAAm, and OXG, and photopolymerized in order to obtain the final TN hydrogels. The evaluation of the TN hydrogels in terms of their mechanical properties using tensile measurements indicated values of fracture stress, fracture strain, and Young’s modulus for the PSAMPS/PAAm/PAAm TN hydrogels similar to those of the conventional PAMPS/PAAm DN hydrogels. In the next report [20], these authors investigated the dependence of the mechanical properties of the PSAMPS/PAAm/PAAm TN hydrogels on several parameters, such as the concentration of the microgel, the concentration of sodium chloride in the second network, the concentration of AAm in the second and third networks, and the concentration of MBAAm in the second network. Increasing the concentration of the microgel or the concentration of AAm in the second and third networks and, therefore, the molar ratio of PAAm to PSAMPS, resulted in great improvement in the mechanical properties of the final TN hydrogels.

#### 2.3.3. TN Hydrogels Containing a Linear Polyelectrolyte Stent as the Second Component

In 2012, the same research group [21] proposed a new method for the toughening of nonionic DN hydrogels by introducing the molecular stent approach. This approach was based on the introduction of a linear polyelectrolyte, PAMPS, between the first and the second polymer networks consisting of nonionic components, in order to induce a higher osmotic pressure, and, consequently a higher degree of swelling in the first network, which would ultimately lead to enhanced mechanical properties. Figure 7 presents the synthetic route followed for the preparation of TN hydrogels based on the molecular stent approach. In order to investigate the universality of this method, several monomers were used in the preparation of the first network, including AAc, AAm, DMAAm, *N*-isopropylacrylamide (NIPAAm), and 2-hydroxyethyl acrylate (HEA), whereas the third network was common for all hydrogels and consisted of loosely cross-linked PAAm polymer chains. The corresponding DN hydrogels without the presence of a PAMPS chain as the second component/stent were also prepared, in order to compare the mechanical properties of the two series of hydrogels. Characterization of all the prepared hydrogels revealed a significant improvement of the tensile mechanical properties of the stent-based TN hydrogels in comparison with the corresponding DN hydrogels lacking the linear PAMPS stent. Furthermore, the tensile mechanical properties of the prepared stent-based TN hydrogels were very similar, and in some cases better, than those of the conventional PAMPS/PAAm DN hydrogels.

One year later, again Gong’s research group [22] developed TN hydrogels consisting of a well-defined first polymer network, a linear PAMPS molecular stent chain as the second component, and a loosely cross-linked nonionic PAAm network as the third network, as shown in Figure 8. The first polymer network was obtained from the reaction of a tetra-amine-terminated four-arm poly(ethylene glycol) (TAPEG) star polymer with an activated tetra-ester-terminated four-arm poly(ethylene glycol) (TNPEG) star polymer. In order to induce a higher osmotic pressure, and, consequently, a higher degree of swelling in the first network which was well-defined, the first network was immersed in an AMPS monomer solution and was UV-irradiated to obtain a DN hydrogel. This DN hydrogel was subsequently immersed in an AAm monomer/MBAAm cross-linker solution, and was also UV-irradiated to prepare the final TN hydrogel. Furthermore, the corresponding PAMPS/PAAm and TPEG/PAAm DN hydrogels were also prepared, in order to perform the comparison between the three types of hydrogels. Characterization of the obtained hydrogels in terms of their mechanical behavior using tensile experiments indicated the superior mechanical properties of the TN hydrogels in comparison with the TPEG/PAAm DN hydrogels without the stent, and the conventional PAMPS/PAAm DN hydrogels. 

#### 2.3.4. TN Hydrogels Prepared Using a Mold

In 2010, the same research group [23] reported the fabrication of TN hydrogels consisting of linear poly(vinyl alcohol) (PVA) as the first component, a highly cross-linked PAMPS second component, and a loosely cross-linked PAAm third network. For comparison, PVA/PAMPS DN and PVA/PAAm DN hydrogels were also prepared. The PVA/PAMPS/PAAm TN hydrogels possessed increased flexibility owing to the highly flexible PVA which acted as an internal mold, and toughness due to the presence of the PAMPS/PAAm DN structure. Figure 9 presents photographs for the PVA/PAMPS/PAAm TN hydrogels which could possess various shapes. Subsequently, the mechanical properties of the prepared PVA/PAMPS DN and PVA/PAAm DN and TN hydrogels were evaluated using tensile measurements. The TN hydrogels exhibited improved mechanical properties in comparison with their DN counterparts, and this was attributed to the presence of the PAMPS/PAAm DN structure.

#### 2.3.5. TN Hydrogels Containing an Electrically Conducting Polymer as the Third Component

In 2009, the research group of Lu [24] were the first to report the preparation of electrically conducting TN hydrogels with good mechanical performance, prepared using the conducting poly(3,4-ethylenedioxythiophene) (PEDOT) as the third component, together with poly(sodium 4-styrenesulfonate) (PNaSS, molecular weight = 70 kg·mol^−1^). In particular, these TN hydrogels consisted of a highly cross-linked PAAc as the first network, a loosely cross-linked PAAc as the second network, and a PEDOT-PNaSS homopolymer mixture as the third component. The polymerization of the 3,4-ethylenedioxythiophene (EDOT) monomer was accomplished through chemical oxidation and was initiated using iron (III) nitrate nonahydrate [Fe(NO_3_)_3_·9H_2_O], while the iron (III) cation (Fe^3+^) also acted as a physical cross-linker for the sodium 4-styrenesulfonate (NaSS) monomer repeating units through ionic interactions, leading to the formation of a physically cross-linked polymer network. In total, five different TN hydrogels were prepared, differing in their PEDOT content with respect to their dry mass, ranging between 8.0 and 18.4 wt.%, while PAAc SN and PAAc/PAAc DN hydrogels were also prepared. The characterization of the conducting TN and their PAAc SN and PAAc/PAAc DN hydrogel precursors in terms of their mechanical performance using both compression and tension experiments indicated higher values of fracture stress and fracture strain for the TN hydrogels in comparison with their SN and DN hydrogel precursors. In addition, the mechanical properties of these TN hydrogels were found to increase upon increasing the PEDOT content, with the TN hydrogel with the highest PEDOT content of 18.4 wt.%, exhibiting a compressive fracture stress of 1.8 MPa and a compressive fracture strain of 78%. Finally, the TN hydrogels exhibited high values of electrical conductivity, up to 10^−3^
*S* cm^−1^, and this value was found to increase when increasing the PEDOT content.

In a later report, Kishi et al. [25] prepared electrically conducting TN hydrogels with satisfactory mechanical properties by introducing a third polymer consisting of PEDOT to the conventional PAMPS/PAAm DN hydrogels. The DN hydrogels were prepared according to Gong’s procedure [19], while the TN hydrogels were prepared in the same manner as previously [24] but using iron (III) *p*-toluenesulfonate hexahydrate [Fe(III) *p*-TS·6H_2_O] as the initiator for the polymerization of the EDOT monomer. Two TN hydrogels with different PEDOT amounts were obtained, and this amount was dependent on the duration of the polymerization of the EDOT monomer. The mechanical properties of the prepared DN and TN hydrogels were evaluated using tensile measurements. The values of fracture stress, fracture strain, and Young’s modulus of the TN hydrogels were higher than the corresponding values of the DN hydrogels. Furthermore, these values were found to improve when the PEDOT amount in the TN hydrogels increased. This increase was attributed to the hydrophobic nature of the EDOT monomer repeating units and the rigid main chain of PEDOT. These TN hydrogels also exhibited electrical conductivity with similar values as in the previous report, in the order of 10^−3^ S cm^−1^, which increased with increasing the PEDOT content.

Two years later, the same research team [26] prepared electrically conducting and mechanically robust TN hydrogels based on poly(styrene sulphonic acid) (PSS), PDMAAm, and PEDOT. In particular, the first network consisted of a highly cross-linked PSS network, the second network contained a loosely cross-linked PDMAAm network, and the third network comprised a PEDOT chain. The TN hydrogels were prepared according to the above-mentioned procedure, using Fe(III) *p*-TS·6H_2_O as the initiator for the polymerization of the EDOT monomer, while the corresponding PSS/DMAAm DN hydrogels were also prepared. Two different DN hydrogels, and, consequently, two different TN hydrogels were obtained, by varying the DMAAm monomer concentration in the second network from 1.5 to 2.0 M, resulting in PEDOT content values in the TN hydrogels equal to 19.4 and 20.3 wt.%, respectively. The characterization of the DN and TN hydrogels in terms of their mechanical properties using compression measurements showed a significant enhancement in fracture stress and Young’s modulus of the TN hydrogels in comparison with their DN hydrogel counterparts, whereas the fracture strain was slightly reduced or remained constant. Furthermore, the values of fracture stress and Young’s modulus of the TN hydrogels were found to increase upon increasing the PEDOT content. Finally, these TN hydrogels exhibited very high values of electrical conductivity, ~1 S cm^‒1^, much higher than those in the previous reports, indicating their great potential for applications as actuators and sensors.

### 2.4. Quadruple-Network (QN) Hydrogels

In 2012, Naficy and co-workers [27] developed electrically conductive, mechanically robust, and pH sensitive TN and QN hydrogels consisting of poly[poly(ethylene glycol methyl ether methacrylate] (PPEGMA), PAAc, and PEDOT-PNaSS (PNaSS: molecular weight = 70 kg·mol^−1^). The first network consisted of a highly cross-linked PPEGMA network, the second network consisted of a loosely cross-linked PAAc network, and the third network consisted of PEDOT-PNaSS physically cross-linked with Fe^3+^. The preparation of the DN and TN hydrogels was accomplished in two or three steps, respectively, while, after the preparation of the TN hydrogels, these were again immersed in the aqueous dispersion containing EDOT monomer and PNaSS and polymerized after reaching swelling equilibrium, which resulted in the formation of the quadruple-network (QN) hydrogels. All the prepared hydrogels were characterized in terms of their mechanical performance using compression and tension measurements. Increasing the amount of the incorporated PEDOT led to a significant increase in the values of both the compressive and tensile fracture stress, whereas the values of compressive fracture strain remained almost constant or were slightly decreased. However, increasing this amount resulted in a significant decrease in the values of the tensile fracture strain. Furthermore, these mechanical properties were found to exhibit a great dependence on pH. Increasing the pH from 3 to 5 led to a great reduction in the values of fracture stress and fracture strain, due to the deprotonation of the carboxylic acid in the AAc monomer repeating units, leading to electrostatic repulsion between them which resulted in higher values of degrees of swelling, and, consequently, poorer mechanical properties. Finally, these QN hydrogels exhibited increased values of electrical conductivity, particularly 4.3 S·cm^−1^, which is the highest reported value among these papers, making these materials potential candidates in soft strain sensors.

In 2016, Shams Es-haghi and Weiss [28] reported the preparation and characterization of nonionic QN hydrogels. These hydrogels were based on AAm monomer and MBAAm cross-linker, while all four PAAm networks were prepared using a relatively high monomer concentration, 4 M, and a very low MBAAm cross-linker concentration (0.01 mol% relative to the monomer), in order to achieve high extensibility and allow for tensile measurements. The thus-prepared SN, DN, TN, and QN hydrogels were characterized in terms of their mechanical properties using both compression and tensile measurements. The tensile measurements indicated high stretch ratios at break for all network multiplicities, but with the SN and DN hydrogels displaying higher stretch ratios at break (1020% and 820%, respectively) as compared to those presented by the TN and QN hydrogels (around 780%). The same tensile measurements also indicated that the fracture stress in tension increased as network multiplicity increased, with values spanning the range from 0.2 MPa for the SN to 1.8 MPa or more for the QN (this sample slipped out of the clamp fixture during the measurement and did not break). A similar trend was observed in the case of the compression experiments. Increasing the network multiplicity led to increased fracture stress values, from ~0.5 MPa for the SN to ~7 MPa for the QN hydrogel, and decreased fracture strain values, from 88% for the SN to 78% for the QN hydrogel. Figure 10 displays the tensile stress-strain curves (Figure 10a) and the compressive stress-strain curves for the SN, DN, TN, and QN hydrogels (Figure 10b). One and two kinks were observed in the curves of the TN and QN hydrogels, respectively, indicating strain localization during tensile deformation. No network damage occurred if the hydrogel was unloaded before the (first) strain localization point. In contrast, large hysteresis was observed if the hydrogel was unloaded after strain localization, manifesting in irreversible energy dissipation arising either from the breaking of cross-linked clusters formed as a result of the high network multiplicity or from the movement of cross-link junctions in the loosely cross-linked networks during deformation.

### 2.5. Quintuple-Network (5×N) Hydrogels

In 2018, our group [29] reported the preparation and study of the compressive mechanical properties of quintuple (5×N) polymer hydrogels (and also their SN, DN, TN and QN precursors) of DMAAm cross-linked with MBAAm, and using five different monomer concentrations, from 1 to 5 M. We found that as network multiplicity and DMAAm monomer concentration increased, fracture stress and Young’s modulus also increased. This resulted in the quintuple network prepared at 5 M DMAAm concentrations being the best-performing multiple polymer hydrogel, displaying a compressive fracture stress and a compressive Young’s modulus of 51 and 2.1 MPa, respectively. Analysis of the fracture stress values recorded for all the prepared multiple network hydrogels, also using the measured swelling degrees, indicated that the improvement of the fracture stress mainly originated from network multiplicity, as the later networks prevent crack formation and propagation in the earlier networks. Subsequently, the concentration of the elastically effective polymer chains in the bulk was calculated using the measured Young’s modulus and the swelling degree values. Very interestingly, this bulk concentration of the elastically effective polymer chains turned out not to change with monomer concentration but to increase linearly with network multiplicity, revealing an increase in strain hardening and/or the concentration of trapped entanglements.

In a very recent study [30], we used nanoindentation to characterize the hardness of some of the above-mentioned [29] multiple network hydrogels. Nanoindentation is a modern method for the characterization of the mechanical properties of even very fragile materials, with the requirement for only a small amount of sample. In these particular experiments, we determined the hardness of the quintuple-network hydrogel prepared at 1 M DMAAm concentrations, and that of its SN, DN, TN and QN precursors. Hardness is the resistance of a material to permanent shape change when a constant compressive force is applied, in the present case, by the nanoindenter. Our measurements indicated that the hardness increased with network multiplicity, arising from the increase in network compactness (decreasing swelling degree) with multiplicity. In addition to hardness, the nanoindentation elastic modulus and the percentage of recoverable energy were also determined, and were both found to increase with network multiplicity as well.

## 3. Applications of the Multiply Interpenetrating Polymer Network Hydrogels

The multiple network hydrogels, e.g., the DN, TN, and QN hydrogels, are excellent candidates in applications that require mechanical strength and toughness. For example, the conventional PAMPS/PAAm DN hydrogels meet these criteria as they possess enhanced mechanical performance, hence they are potential candidates in applications that require both soft and wet materials, such as soft robotics, including artificial articular cartilages and artificial tendons [19]. In addition, in order for the hydrogels to be able to serve in applications that include motion, the hydrogels must also exhibit low frictional coefficients, such as the PAMPS/PAAm/PAMPS TN hydrogels [12].

Furthermore, the DN, TN, and QN hydrogels with increased mechanical properties can serve as scaffolds in tissue engineering, due to their biocompatibility and non-cytotoxicity. For example, TN hydrogels based on DMAAm and SAMPS [13] with satisfactory mechanical performance, exhibited the ability to induce cell spreading and proliferation for particular types of cells. However, the hydrogels must also be biodegradable in order to enable their clearance from the body.

Injectable TN microgels composed of Odex, Teleostean, and CEC [14] may be applied in biomedical applications such as drug delivery, because these materials have the appropriate size to serve as injectable materials, and, at the same time, they can be degraded once they are administered in the body owing to their biodegradable constituents. Finally, the TN hydrogels containing an electron conducting component, e.g., PEDOT, in their structure and exhibit satisfactorily enhanced mechanical properties may be employed in the technology field, in applications such as sensing and actuating [24,25,26,27].

## 4. Conclusions

We have reviewed the small, but growing, literature on multiple interpenetrating polymer network hydrogels, consisting of three or more polymer networks/components. Mostly, TN hydrogels of various monomer types and concentrations and different cross-linking densities have been prepared and characterized. The incorporation of an extra polymer network/component into a DN hydrogel leads to enhancement of the mechanical properties of the resulting TN hydrogel. Increasing the polymer volume fraction of the loosely cross-linked polymer networks appears to induce an increased protection towards fracture, as the higher polymer networks serve as soft matrix, facilitating energy dissipation and protecting the hydrogel from failure. A key parameter for the improvement of mechanical properties is the molar ratio of the monomer repeating units in the higher networks to the monomer repeating units in the first network. Thus, increasing network multiplicity and/or monomer concentration in the higher networks results in a higher molar ratio, thereby enhancing the mechanical properties, as was the case for the QN and 5×N hydrogels. Furthermore, using a linear polyelectrolyte as stent between two nonionic polymer networks results in a higher osmotic pressure of the first network, leading to a higher degree of swelling, and, consequently, to increased mechanical performance. Such mechanically robust hydrogels can find application as artificial tissues; however, these applications also require low frictional coefficients, which can be obtained by using a charged polymer network as the third component. In addition, when the TN hydrogels consist of biodegradable and biocompatible materials, these hydrogels can serve as scaffolds for tissue engineering. Finally, when the TN hydrogels consist of an electrically conducting polymer as the third polymeric component, the TN hydrogels exhibit high values of electrical conductivity, in addition to the enhanced mechanical properties, making these materials ideal for use in sensors.

The modularity in the procedure for the preparation of multiple network hydrogels makes their synthesis very easy, and the addition of extra functionalities very simple. This would be particularly useful for the incorporation of new functions from the rapidly growing polymer hydrogel literature, including, but not limited to, elements that further improve network mechanical properties [32,33,34,35,36,37]. Thus, we predict a bright future for this field, where a variety of carefully chosen properties are introduced, so that these materials perfectly match an intended application. 

## Figures and Tables

**Figure 1 gels-05-00036-f001:**
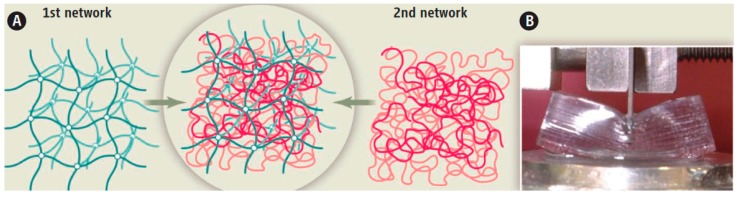
(**Α**) Schematic representation of a double-network (DN) hydrogel prepared from the combination of two different polymer networks. (**Β**) Photograph of a tough DN hydrogel containing 90 wt% water. Adapted and reprinted with permission from [11]. Copyright 2014 American Association for the Advancement of Science.

**Figure 2 gels-05-00036-f002:**
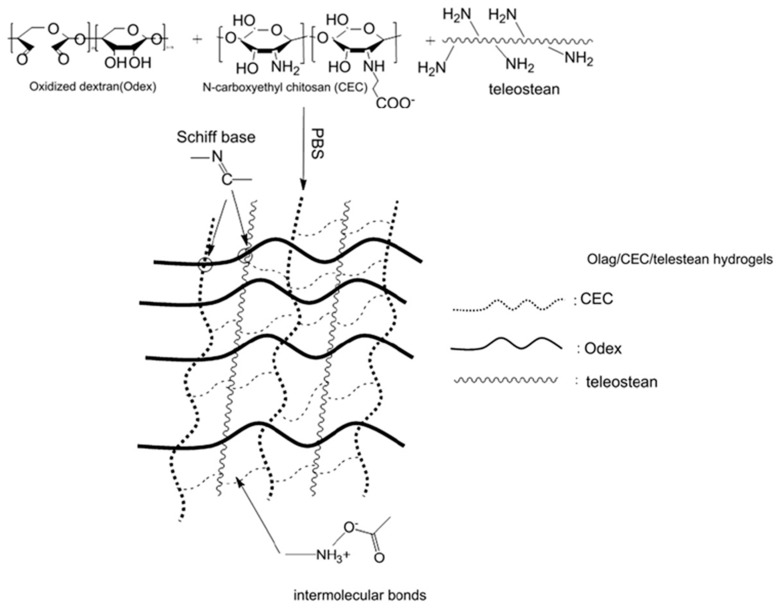
Structure of the three polymeric components used to prepare the triple-network (TN) hydrogels. Adapted and reprinted with permission from [14]. Copyright 2011 Elsevier.

**Figure 3 gels-05-00036-f003:**
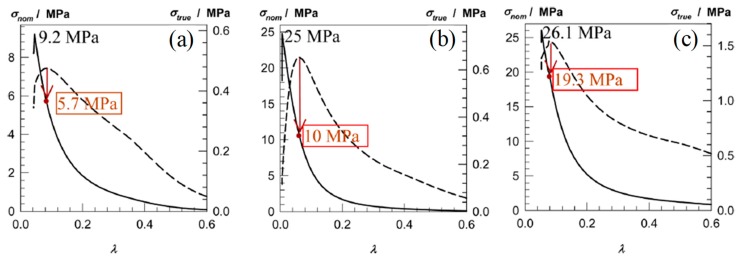
Typical compressive stress–strain curves for the hydrogels, plotting both the nominal (solid curves) and true (dashed curves) stress against the deformation ratio, λ. Red circles are taken as the points of failure in the gel samples. (**a**) PAAm/PAAm double-network (DN) hydrogel formed at 4 mol% PEGDMA. *n*_21_ = 3.6. (**b**) PAAm/PAAm/PAAm triple-network (TN) hydrogel formed at 4 mol% PEGDMA. *n*_21_ = 2.6, *n*_32/1_ = 17. (**c**) PDMAAm/PDMAAm/PDMAAm TN hydrogel prepared using 10 mol% PEGDMA. *n*_21_ = 4.0, *n*_32/1_ = 33. PAAm: polyacrylamide; PDMAAm: poly(*N*,*N*-dimethylacrylamide). Adapted and reprinted with permission from [16]. Copyright 2014 American Chemical Society.

**Figure 4 gels-05-00036-f004:**
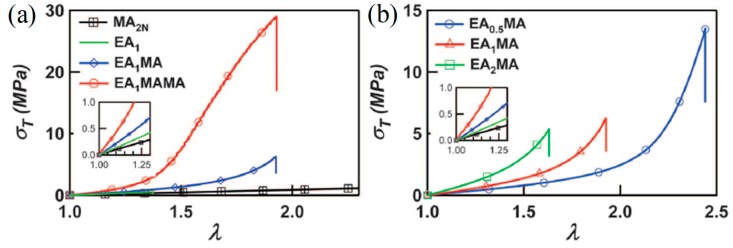
Mechanical behavior of the single network (SN), double-network (DN) and triple-network (TN) elastomers. (**a**) True stress–stretch curves for the EA_1_, EA_1_MA, EA_1_MAMA and PMA second network alone at 60 °C. (**b)** Effect of the cross-linker loading in the EA first network on the EA_x_MA DN elastomers. EA: ethyl acrylate; MA: methyl acrylate. Adapted and reprinted with permission from [17]. Copyright 2014 American Association for the Advancement of Science.

**Figure 5 gels-05-00036-f005:**
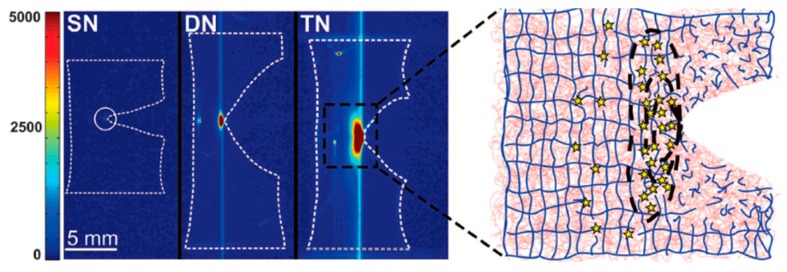
Mapping of location where bonds break during crack propagation. (**Left**) Intensity-colored images of propagating cracks on notched samples containing a chemiluminescence cross-linker in the first network, showing light emission due to breaking of bonds in the single-network (SN), double-network (DN), and triple-network (TN) samples. (**Right**) Schematic of the sacrificial bond-breaking mechanism in front of the crack tip for the DN and TN; the first network is represented in blue, and the second and third networks are in red. Adapted and reprinted with permission from [17]. Copyright 2014 American Association for the Advancement of Science.

**Figure 6 gels-05-00036-f006:**
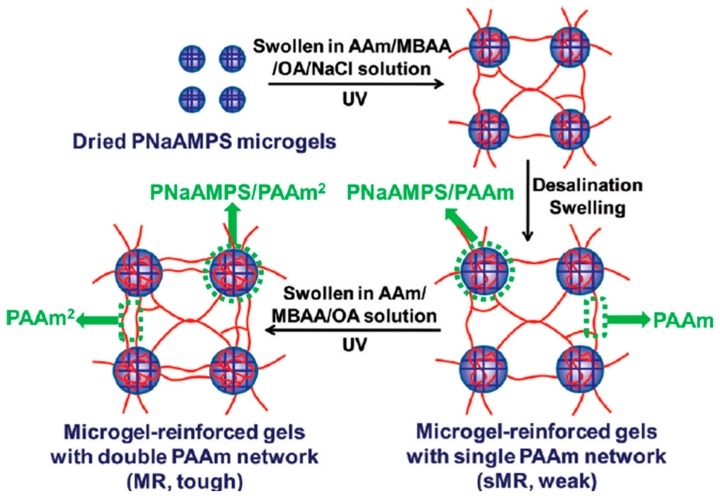
Preparation route followed for the synthesis of the PAMPS/PAAm/PAAm triple-network (TN) hydrogels. PAMPS: poly(2-acrylamido-2-methylpropane sulfonic acid); PAAm: polyacrylamide. Adapted and reprinted with permission from [20]. Copyright 2012 American Chemical Society.

**Figure 7 gels-05-00036-f007:**
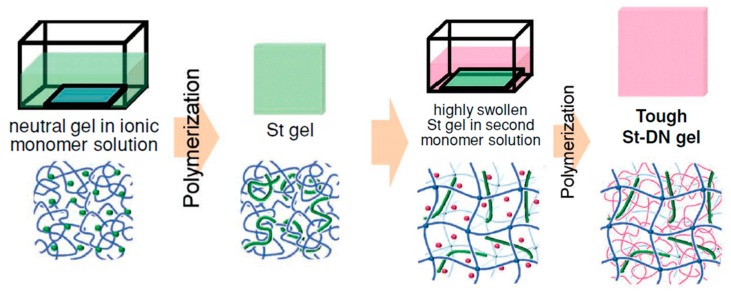
Schematic representation of the preparation of the triple-network (TN) hydrogels containing a polyelectrolyte molecular stent as the second component. After its preparation, the first nonionic polymer network was immersed in the ionic 2-acrylamido-2-methylpropane sulfonic acid (AMPS) monomer solution, and, after its equilibrium swelling, the resulting composite system was photopolymerized in order to obtain the double-network (DN) hydrogel. Then, the DN hydrogel was immersed in an aqueous solution of the second acrylamide (AAm) monomer, and this system was photopolymerized in order to obtain the TN hydrogel. Adapted and reprinted with permission from [21]. Copyright 2012 John Wiley & Sons, Inc.

**Figure 8 gels-05-00036-f008:**
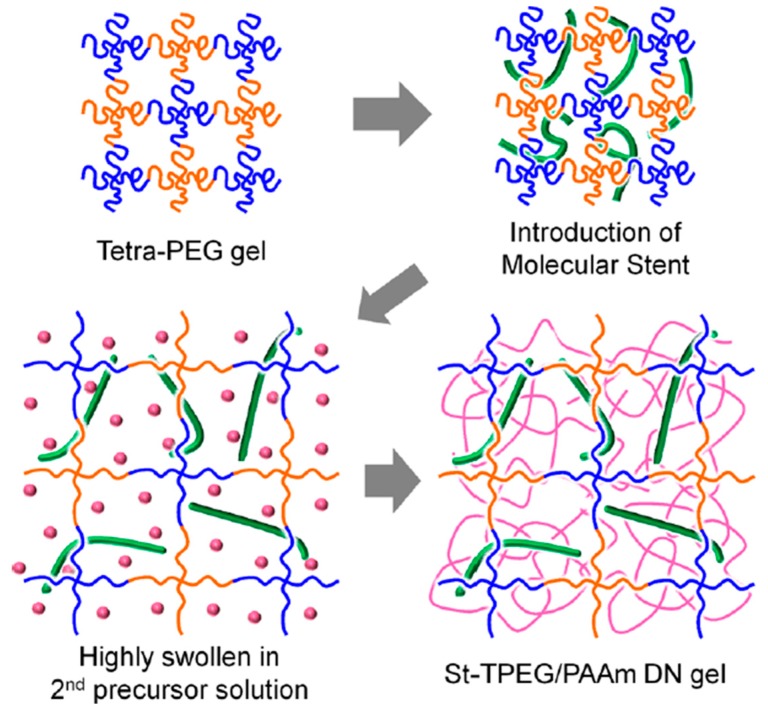
Schematic representation of the procedure followed for the preparation of the triple-network (TN) hydrogels based on a well-defined four-arm poly(ethylene glycol) star (tetraPEG) gel first polymer network. After its preparation, the first network was immersed in the ionic 2-acrylamido-2-methylpropane sulfonic acid (AMPS) monomer solution, and after its photopolymerization, the resulting double-network (DN) hydrogel was immersed in an aqueous acrylamide (AAm)/*N*,*Nˊ*-methylenebisacrylamide (MBAAm) solution and photopolymerized to prepare the final TN hydrogel. Adapted and reprinted with permission from [22]. Copyright 2013 ACS Publications American Chemical Society.

**Figure 9 gels-05-00036-f009:**
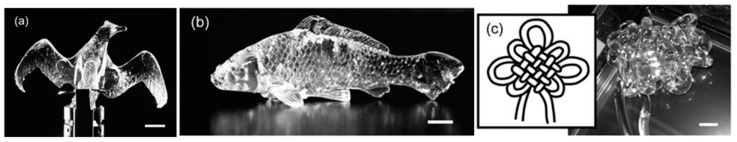
Pictures for the PVA/PAMPS/PAAm triple-network (TN) hydrogels with the shape of a (**a**) bird, (**b**) fish, and (**c**) Chinese knot. Scale bars: 1 cm. PVA: poly(vinyl alcohol); PAMPS: poly(2-acrylamido-2-methylpropane sulfonic acid); PAAm: polyacrylamide. Adapted and reprinted with permission from [23]. Copyright 2010 Royal Society of Chemistry.

**Figure 10 gels-05-00036-f010:**
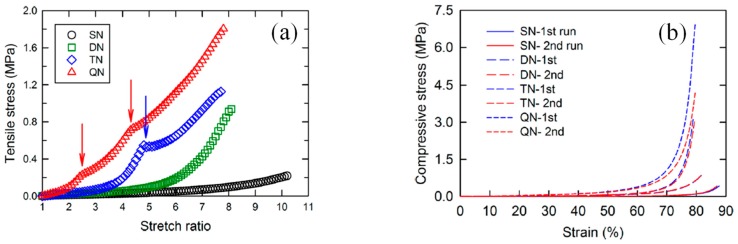
(**a**) Tensile stress–strain curves for the single-network (SN), double-network (DN), triple-network (TN), and quadruple-network (QN) hydrogels. Arrows show the strain localization points in the TN and QN hydrogels. (**b**) Compressive stress–strain curves for the SN, DN, TN, and QN hydrogels. Blue lines show the first compression run on the sample, and the red lines show the second compression after the first run on the same sample. Adapted and reprinted with permission from [28]. Copyright 2016 American Chemical Society.

**Table 1 gels-05-00036-t001:** Summary of all prepared triple-network (TN) hydrogels and their mechanical properties, fracture stress (σ_max_), fracture strain (ε_max_), and Young’s modulus (*Ε*), in comparison with the mechanical properties of their single-network (SN) and double-network (DN) precursors.

Gel Components			Method	Mechanical Properties of SN			Mechanical Properties of DN			Mechanical Properties of TN			Ref.
1st Network	2nd Network	3rd Network		*σ_max_* (MPa)	*ε_max_* (%)	*E* (MPa)	*σ_max_* (MPa)	*ε_max_* (%)	*E* (MPa)	*σ_max_* (MPa)	*ε_max_* (%)	*E* (MPa)	
**(a) Simple Triple Networks**													
AMPS-1 MBAAm-8	AAm-2-MBAAm-0.1	AMPS-1 MBAAm-0.1	Compression	-	-	-	4.60	65	0.84	4.80	57	2.00	[12]
		AMPS-1 MBAAm-0								9.20	70	2.10	
SAMPS-1 MBAAm-4	DMAAm-3 MBAAm-0.1	SAMPS-*co*-DMAAm (F = 0.5)-1 MBAAm-0	Compression	0.26	57	0.19	-	-	-	3.00	71	0.32	[13]
		SAMPS-*co*-DMAAm (F = 0.5)-1 MBAAm-2					-	-	-	2.31	65	0.65	
		SAMPS-*co*-DMAAm (F = 0.5)-1 MBAAm-4					-	-	-	1.36	47	0.88	
Odex	Teleostean	CEC	-	-	-	-	-	-	-	-	-	-	[14,15]
AAm-1.40 PEGDMA-4	AAm-0.7–7.0 MBAAm-0	AAm-0.7–7.0 MBAAm 0	Compression	0.15	70	-	9.20	95	-	25.00	99	0.10	[16]
DMAAm-1 PEGDMA-4–10	DMAAm-0.5–5.0 MBAAm-0	DMAAm-0.5–5.0 MBAAm-0	Compression	0.17	47	-	-	-	-	26.10	95	2.00	
EA-5 BDA-1.45	EA-9.4 BDA-0.01	EA-9.4 BDA-0.01	Tension	1.20	-	0.60	10.00	260	0.80	16.00	220	1.50	[17]
EA-5 BDA-1.45	MA-11 BDA-0.01	MA-11 BDA-0.01		0.50	-	0.80	8.00	240	1.30	22.00	260	2.20	
EA-5 BDA-2.81				0.50	-	1.50	6.50	190	2.00	29.00	190	4.20	
EA-5 BDA-5.81				0.50	-	2.30	3.00	160	2.30	-	-	-	
MHA (20 g·L^−1^)	DMAAm-3 MBAAm-0.05	DMAAm-3 MBAAm-0.05	Compression	0.05	40	0.017	12.00	93	0.37	22.00	96	0.40	[18]
M_i_-x_i_, C_i_-y_i_: M_i_, x_i_, C_i_, and y_i_ state the abbreviation of the polymer’s name, the molar concentration of monomer, the abbreviation of the cross-linker’s name, and the cross-linker loading feed in mol% with respect to the monomer, respectively. PC: physically cross-linked.													
**(b) Triple Networks with a 1st Polymer Network Based on Microgels**													
SAMPS-1 MBAAm-4	AAm-2 MBAAm-0.01	AAm-4 MBAAm-0.01	Compression	-	-	-	0.15	130	0.05	2.46	1270	0.22	[19,20]
AAm-1-MBAAm-4				-	-	-	0.41	1010	0.03	1.37	910	0.05	[20]
DMAPAA-Q-1 MBAAm-4				-	-	-	-	-	-	0.94	970	0.15	
NaSS-*co*-DMAEA-Q (F = 0.5)-1-MBAAm-4				-	-	-	-	-	-	0.75	530	0.07	
SAMPS+DMAPAA-Q (F = 0.5)-1 MBAAm-4				-	-	-	-	-	-	0.51	410	0.07	
**(c) Triple Networks with a Linear Polyelectrolyte Stent as the 2nd Polymeric Component**													
AAm-1.2 MBAAm-4	AMPS-1 MBAAm-0	AAm-2 MBAAm-0.02	Tension	-	-	-	-	-	-	0.83	1000	-	[21]
DMAAm-0.7 MBAAm-3				-	-	-	-	-	-	1.95	-	-	
DMAAm-1 MBAAm-2				-	-	-	-	-	-	1.57	-	-	
NIPAAm-0.7 MBAAm-2				-	-	-	-	-	-	1.02	-	-	
AAc-1 MBAAm-4	AMPS-0.7 MBAAm-0	AAm-2 MBAAm-0.02	Tension	-	-	-	-	-	0.067	0.70	-	-	
AAm-1 MBAAm-4				0.031	-	-	-	-	0.15	0.69	-	-	
HEA-1 MBAAm-4				0.037	-	0.054	-	-	-	0.82	-	-	
	NaSS-1 MBAAm-0						-	-	-	0.34	-	-	
	DMAPAA-Q-1 MBAAm-0						-	-	-	0.47	-	-	
	DMAEA-1 MBAAm-0						-	-	-	0.47	-	-	
TPEG-2 × 10^−5^	AMPS-0.6 MBAAm-0		Tension	-	-	-	0.20	600	-	2	2200	-	[22]
	AMPS-1.0 MBAAm-0									-	-	-	
**(d) Triple Networks with a Mold Used in the 1st Polymeric Component**													
PVA-PC	AMPS-1 MBAAm-4	AAm-2-MBAAm-0	Tension	-	-	-	0.30	300	-	0.80	900	-	[23]
**(e) Triple Networks Containing an Electrically Conducting Polymer as the 3rd Component**													
AAc-1.5 MBAAm-6	AAc-6 MBAAm-0.1	EDOT (0.48 M)-PNaSS (0.10 M) PC	Tension	0.10	37	-	0.60	61	-	1.00	68	-	[24]
		EDOT-PNaSS (No. 2) PC								1.10	70	-	
		EDOT-PNaSS (No. 3) PC								1.30	72	-	
		EDOT-PNaSS (No. 4) PC					0.60	61	-	1.60	73	-	
		EDOT-NaSS (No. 5) PC								1.80	78	-	
AMPS-1 MBAAm-4	AAm-2 MBAAm-0.1	EDOT (No. 1)	Tension	-	-	-	1.19	134	0.37	1.38	154	0.33	[25]
		EDOT (No. 2)								2.07	235	0.56	
NaSS-1 MBAAm-10	DMAAm-1.5 MBAAm-0	PEDOT	Compression	-	-	-	0.39	45	0.71	1.27	45	3.48	[26]
	DMAAm-2.0 MBAAm-0						1.08	81	0.95	1.98	76	2.97	
PEGMA-0.18 MBAAm-4	AAc-2.78 MBAAm-0.1	EDOT (0.46 M) + NaPSS (0.48 M) PC	Tension	-	-	-	0.48	340	0.40	0.60	240	0.11	[27]
			Compression				8.50	81	-	11.60	78	-	

**Table 2 gels-05-00036-t002:** Chemical structures and names of monomers and polymers employed in the preparation of the triple-network (TN) hydrogels.

Chemical Structure	Name
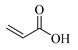	AAc, Acrylic acid
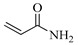	AAm, Acrylamide
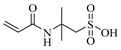	AMPS, 2-Acrylamido-2-methylpropane sulfonic acid
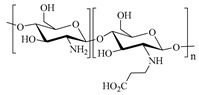	CEC, *Ν*-Carboxyethyl chitosan
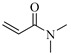	DMAAm, *Ν*,*Ν*-Dimethylacrylamide
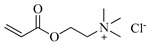	DMAEA-Q, 2-(Trimethylamino)ethyl acrylate, chloride quaternary salt
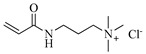	DMAPAA-Q, 3-(Acrylamidopropyl)trimethylammonium chloride
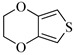	EDOT, 3,4-Ethylenedioxythiophene
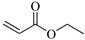	EΑ, Ethyl acrylate
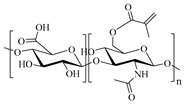	MHA, Methacrylated hyaluronan
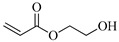	HEA, 2-Hydroxyethyl acrylate
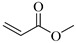	MΑ, Methyl acrylate
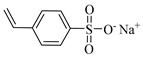	NaSS, Sodium 4-styrenesulfonate
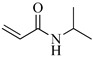	NIPAAm, *N*-Isopropylacrylamide
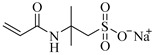	SΑMPS, 2-Acrylamido-2-methyl-propane sulfonic acid sodium salt
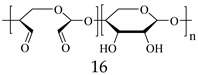	ODEX, Partially oxidized dextran
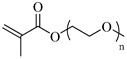	PEGMA, Poly(ethylene glycol) methyl ether methacrylate
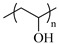	PVA, Poly(vinyl alcohol)

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
