# Peer review of "Multiply Interpenetrating Polymer Networks: Preparation, Mechanical Properties, and Applications"

_gels, 2019, doi:10.3390/gels5030036_

Round 1
Reviewer 1 Report
Patrickios and Panteli summarized the recent progress in multiple IPN hydrogels in this review. It covers most of the work on “double-network hydrogels”, a term invented by Gong and Osada, which has been an important branch of the synthetic hydrogel family. Lots of works, mainly, from Gong and Osada, as well as other group like Okay, have been reviewed and discussed here. It can be a good guidance for readers who are willing to understand how this field is progressing, however, before the publication, I highly recommend the authors to address these concerns:
1. Regarding double network, Gong and Osada have defined the term very clearly in the Adv Mater work in 2003, which will not bring any doubt when people are using this term. However, when mentioning triple, quadruple or even quintuple network, is there any specific definition or requirement on each network? Similar to DN, Gong and Osada have well explained the difference between double network and interpenetrating polymer network, can the author define, at least specifically point out the criteria for a triple, quadruple or even quintuple network. It can not be simply just referred to be a network prepared through multiple steps, involving multiple components.
2. Since the reviewer cite the work from Creton (ref 17), the review doubts that if this work should be an example for this triple hydrogel network? If a triple network can cover elastomer, does it mean all those dry-state plastic/elastomer compounds should also be considered?
3. The author mentioned the “preparation, mechanical properties and application”, however, from the reviewer’s viewpoint, this review are specifically focused on preparation. It should be good for the author to organize a specific section to “mechanical properties” on how to understand in a triple, quadruple or even quintuple network, how each network contribute to the mechanics of the overall network, and how they synergistically interact and decide the macroscopic mechanical performance. The same as “application”, Gong has done some work on developing such double network as structure biomaterials, interfacial adhesion, etc, such work should be summarized in the specific section of “application”.
4. In the introduction, when the reviewer mentioning that “In the last two decades, several new hydrogel structures possessing improved mechanical properties have been reported.” The following three recently published related important papers are recommended to be cited properly here to support this statement: Adv. Mater. 2017, 29, 1605325; Adv. Mater., 2017, 29, 1604951 and Advanced Materials, 2018, 30, 1707169.
Author Response
Comments:
Patrickios and Panteli summarized the recent progress in multiple IPN hydrogels in this review. It covers most of the work on “double-network hydrogels”, a term invented by Gong and Osada, which has been an important branch of the synthetic hydrogel family. Lots of works, mainly, from Gong and Osada, as well as other groups like Okay, have been reviewed and discussed here. It can be a good guidance for readers who are willing to understand how this field is progressing, however, before the publication, I highly recommend the authors to address these concerns:
Our Response: We thank the Referee for his/her good words regarding the width and usefulness of our review, and below we try to address all of his/her concerns.
1. Regarding double network, Gong and Osada have defined the term very clearly in the Adv Mater work in 2003, which will not bring any doubt when people are using this term. However, when mentioning triple, quadruple or even quintuple network, is there any specific definition or requirement on each network? Similar to DN, Gong and Osada have well explained the difference between double network and interpenetrating polymer network, can the author define, at least specifically point out the criteria for a triple, quadruple or even quintuple network. It cannot be simply just referred to be a network prepared through multiple steps, involving multiple components.
Our Response: This is an excellent and extremely insightful comment by this Reviewer. Indeed, the presently reviewed mechanically-enhanced multiple networks are believed to be more than simply interpenetrated, and not involving any attractive or repulsive interactions among their components. Similar to the DN concept, the second network, and, in general, each subsequent higher network, should be more fluid (i.e., less viscous, and more ductile) than the previous ones, so as to dissipate mechanical energy and prevent crack formation and crack propagation in the previous networks. This may suggest an optimal multiple network design to be tested in the future, in which each subsequent network comprises a gradually higher monomer concentration and a gradually lower cross-linker concentration. This clarifications and comments have now been added towards the end of page 2 of the revised manuscript.
2. Since the reviewer cite the work from Creton (ref 17), the review doubts that if this work should be an example for this triple hydrogel network? If a triple network can cover elastomer, does it mean all those dry-state plastic/elastomer compounds should also be considered?
Our Response: The Referee disputes here whether the work by Creton and co-workers should be included in our review, because the materials developed by these researchers are elastomers rather than hydrogels. We recognize the Referee’s point, but we believe that the work is very nice and it is worth to be incorporated in our article. Furthermore, this design appears to fulfill the criteria set out in the original 2003 DN paper in Advanced Materials by Gong and Osada (see previous comment), in which the higher networks indeed comprise more monomer and less cross-linker, so as to more efficiently dissipate mechanical energy and toughen the final material. Finally, it is nice to see that the DN and multiple network concepts can also be generalized and applied for systems in organic solvents (organogels) and in the bulk (elastomers).
3. The author mentioned the “preparation, mechanical properties and application”, however, from the reviewer’s viewpoint, this review is specifically focused on preparation. It should be good for the author to organize a specific section to “mechanical properties” on how to understand in a triple, quadruple or even quintuple network, how each network contributes to the mechanics of the overall network, and how they synergistically interact and decide the macroscopic mechanical performance. The same as “application”, Gong has done some work on developing such double network as structure biomaterials, interfacial adhesion, etc, such work should be summarized in the specific section of “application”.
Our Response: The suggestion by the Reviewer on a dedicated “mechanical properties” section is an insightful one too. To follow this suggestion, we carefully went through all articles on multiple networks summarized and cited in our review. In doing so, however, we could not find a new mechanism put forward by the authors of those articles to explain the further enhancement of mechanical properties, other than the mechanism proposed to explain the improved mechanical properties of classical double-network hydrogels. Thus, at this instance, we do not change the structure of our Review. Regarding the second point of the Reviewer within this comment, about more “applications” to be mentioned, the Referee is correct in that Gong has published several works on mechanically robust hydrogels as biomaterials, but these concern double-network hydrogels, which need not be mentioned in our review specializing on triply or higher interpenetrating polymer networks.
4. In the introduction, when the reviewer mentioning that “In the last two decades, several new hydrogel structures possessing improved mechanical properties have been reported.” The following three recently published related important papers are recommended to be cited properly here to support this statement: Adv. Mater. 2017, 29, 1605325; Adv. Mater., 2017, 29, 1604951 and Adv. Mater. 2018, 30, 1707169.
Our Response: We gladly cite the recommended references in the Conclusions and Outlook section of our revised manuscript, and thank the Reviewer for pointing them out to us. However, in our original statement, we only meant hydrogel structures developed in 2001 – 2003, i.e., sliding ring gels, nanocomposite gels, and double-network gels.
Reviewer 2 Report
The authors review the recent progress on the multiple network hydrogels with remarkable mechanical toughness. The research on the multiple-network tough hydrogels started with double network hydrogels, and then has been extended to triple, quadruple, and quintuple network hydrogels with additional functions. This paper is useful to overview the rapidly growing research field of multiple tough hydrogels, but some points should be clarified for the publication:- In addition to the multiple-network hydrogels, other toughening mechanisms for hydrogels have been reported. For example, introducing reversible weak cross-links enhances mechanical toughness of hydrogels with self-recovery and/or self-healing properties [Sun, J. et al., “Highly stretchable and tough hydrogels”, Nature, 489(7414), 133 (2012)]. The toughness of the self-healing hydrogels is comparable to that of the multiple-network hydrogels [Li, J. et al., “Hybrid hydrogels with extremely high stiffness and toughness”, ACS Macro Letters, 3(6), 520-523 (2014)]. What are advantages and disadvantages of the multiple-network hydrogels compared with other tough hydro gels including the self-healing tough hydrogels with reversible crosslinks ?
- Table 1 shows various monomers used for the multiple-network hydrogels. Does the mechanical properties of the hydrogels depend on type of monomer? Does the chemical interaction between interpenetrated networks influence their mechanical properties ?
Author Response
The authors review the recent progress on the multiple network hydrogels with remarkable mechanical toughness. The research on the multiple-network tough hydrogels started with double network hydrogels, and then has been extended to triple, quadruple, and quintuple network hydrogels with additional functions. This paper is useful to overview the rapidly growing research field of multiple tough hydrogels, but some points should be clarified for the publication:
Our Response: We thank this Referee too for also pointing out the usefulness of our review, and in the following we clarify the points he/she raises.
1. In addition to the multiple-network hydrogels, other toughening mechanisms for hydrogels have been reported. For example, introducing reversible weak cross-links enhances mechanical toughness of hydrogels with self-recovery and/or self-healing properties [Sun, J. et al., “Highly stretchable and tough hydrogels”, Nature, 489(7414), 133 (2012)]. The toughness of the self-healing hydrogels is comparable to that of the multiple-network hydrogels [Li, J. et al., “Hybrid hydrogels with extremely high stiffness and toughness”, ACS Macro Letters, 3(6), 520-523 (2014)]. What are advantages and disadvantages of the multiple-network hydrogels compared with other tough hydrogels including the self-healing tough hydrogels with reversible crosslinks?
Our Response: This is a very good point brought up by the Reviewer. A clear disadvantage of the double and multiple network hydrogels is their lack of fatigue resistance, as they cannot self-repair after failure. On the other hand, although self-healing hydrogels do possess fatigue resistance, the chemistry of these materials is a bit restrictive, in that they must comprise many weak associations. Furthermore, the multiplicity of these associations generally leads to a lower water content (60-70%) as compared to the double networks (80-90%). These suggested references have been added in the Conclusions and Outlook section of the revised manuscript, together with some appropriate comments.
2. Table 1 shows various monomers used for the multiple-network hydrogels. Does the mechanical properties of the hydrogels depend on type of monomer? Does the chemical interaction between interpenetrated networks influence their mechanical properties?
Our Response: This is another good point by this Referee. Monomer type influences the mechanical properties only indirectly, via the swelling degrees, which affect the ratio of the polymer volume fractions of the networks. This information has been added at the beginning of page 2 of the revised manuscript. Regarding chemical interactions between (among) the interpenetrating network components, these are not desirable, as the envisioned toughening mechanism in the double (and multiple) network hydrogels is mechanical energy dissipation by a dense and fluid second (or higher) network to prevent crack formation and propagation in the previous networks (see our response to Comment no. 1 of Reviewer no. 1).
Reviewer 3 Report
This is a very good review on triple network hydrogels. I think that the authors correctly responded the comments of the reviewer except the 2nd one. For instance, the first comment of the reviewer, namely “can the author define, at least specifically point out the criteria for a triple, quadruple or even quintuple network?” already responded on page 2, bottom paragraphs.
However, the response to the second comment is inadequate. Although the work of Creton (ref 17) dealing with the elastomers should be cited in this review, this is NOT because “it is a very nice and worth to be incorporated in the article”. The reason is the fact that the idea of triple networking came from Creton (ref 17) due to the following reason:
The triple networking strategy was developed by Okay et al (reference 16) to overcome one of the limitations of the classical DN technique. Because the swelling ratio of hydrogels is inversely proportional to the cross-link density, the DN technique is limited to the use of polyelectrolyte first networks, which hinders its many applications. Triple-network approach bases on the loss of the translational entropy of a second monomer upon its polymerization within the first network. As noted in ref. 16, the idea of using the entropy loss on polymerization of a second monomer to permit swelling by a third monomer has been published by Creton (reference 17) in bulk elastomers rather than hydrogels. As mentioned in ref 16, “the entropy of second monomer, if polymerized in a first network hydrogel, will decrease so that additional solvent (third monomer) can enter into the gel phase to assume its new thermodynamic equilibrium. This means that, assuming the cross-link density of the first network does not change much after DN formation and both networks consist of the same polymer, DN will swell more than the first network so that triple networks could be prepared.”
The authors should take into account these important points and revise the manuscript accordingly.
Author Response
This is a very good review on triple network hydrogels. I think that the authors correctly responded the comments of the reviewer except the 2nd one. For instance, the first comment of the reviewer, namely “can the author define, at least specifically point out the criteria for a triple, quadruple or even quintuple network?” already responded on page 2, bottom paragraphs. However, the response to the second comment is inadequate. Although the work of Creton (ref 17) dealing with the elastomers should be cited in this review, this is NOT because “it is a very nice and worth to be incorporated in the article”. The reason is the fact that the idea of triple networking came from Creton (ref 17) due to the following reason:
The triple networking strategy was developed by Okay et al (reference 16) to overcome one of the limitations of the classical DN technique. Because the swelling ratio of hydrogels is inversely proportional to the cross-link density, the DN technique is limited to the use of polyelectrolyte first networks, which hinders its many applications. Triple-network approach bases on the loss of the translational entropy of a second monomer upon its polymerization within the first network. As noted in ref. 16, the idea of using the entropy loss on polymerization of a second monomer to permit swelling by a third monomer has been published by Creton (reference 17) in bulk elastomers rather than hydrogels. As mentioned in ref 16, “the entropy of second monomer, if polymerized in a first network hydrogel, will decrease so that additional solvent (third monomer) can enter into the gel phase to assume its new thermodynamic equilibrium. This means that, assuming the cross-link density of the first network does not change much after DN formation and both networks consist of the same polymer, DN will swell more than the first network so that triple networks could be prepared.”
Our Response: We thank the Referee for his/her nice words about our review paper, and also his/her comments for improving the quality of our review. Following his/her suggestion, we have kept in the re-revised manuscript the Science paper by Creton and co-workers, and also added the justification for its importance (last paragraph on page 3 of the manuscript).
The authors should take into account these important points and revise the manuscript accordingly.
Our Response: We have done so!
Round 2
Reviewer 1 Report
The reviewer really thinks the authors need to take into consideration the reviewers' comments for their correction, since these are based on the reviewers' understanding and can really leverage the quantity and impact of this review.
"Double network" has been clearly defined by Gong and coworkers how is the composition and characteristic for each component. Unfortunately, the authors still stubbornly includes the work from Creton. The reviewer personally agrees that this is a great work, however, it is not a hydrogel system, it is within the scope of DN. If not, why the authors did cover those IPN-based elastomer or plastic composite systems inside?
Gong has clearly established the mechanism behind the toughening effect of double network in her tutorial review in Soft Matter "2010, 6, 2583", and the authors really needs to spend some effort to catch the mechanisms and write a separate section on the mechanics of multiple networks.
Author Response
The reviewer really thinks the authors need to take into consideration the reviewers' comments for their correction, since these are based on the reviewers' understanding and can really leverage the quantity and impact of this review.
Our Response: We thank the Reviewer for his/her comments, aiming at improving the quality and appeal of our manuscript. We have now added a new section in the re-revised manuscript on the mechanism of mechanical property enhancement and mechanism of fracture of double-network hydrogels (new section “2.2 Toughening Mechanism in DN Hydrogels”), as found in Gong’s tutorial review in Soft Matter (2010, 6, 2583) that this Reviewer suggested. However, we have not taken away from the re-revised manuscript the information on Creton’s paper because the adjudicative Reviewer stated that it is important that the paper be kept in the manuscript, and gave a detailed justification for that.
1. "Double network" has been clearly defined by Gong and coworkers how is the composition and characteristic for each component. Unfortunately, the authors still stubbornly include the work from Creton. The reviewer personally agrees that this is a great work, however, it is not a hydrogel system, it is within the scope of DN. If not, why the authors did cover those IPN-based elastomer or plastic composite systems inside?
Our Response:
Regarding the first point in this comment, we had already added in the previously revised manuscript the characteristics of the two components in the DNs.
Regarding the second point in this comment, we would have gladly removed the work of Creton in our manuscript, given the insistent of this Reviewer. However, given the statement of the other (adjudicative) Reviewer about the importance of that work, accompanied with a detailed justification, we decided to keep it.
2. Gong has clearly established the mechanism behind the toughening effect of double network in her tutorial review in Soft Matter "2010, 6, 2583", and the authors really needs to spend some effort to catch the mechanisms and write a separate section on the mechanics of multiple networks.
Our Response: We thank the Referee for his/her suggestion. We have now carefully read the particular tutorial review, and, based on that, we have added new section “2.2 Toughening Mechanism in DN Hydrogels”, and also added a paragraph in (new) section “2.3 Triple-Network Hydrogels”, providing the possible mechanism of mechanical property enhancement in TN and multiple network hydrogels as well.